# UNLEASH MODEL CAPACITY FOR UNIVERSAL DENSE RETRIEVAL BY TASK SPECIALTY OPTIMIZATION

## ABSTRACT

Universal dense retrieval, with one unified representation space to empower various retrieval scenarios, has many appealing advantages in simplicity, efficiency, and potential to break echo chambers with cross-scenario information access. However, standard multi-task trained dense retrievers often fail to meet the accuracy of scenario-specific models. In this paper, we analyze the multi-task learning in universal retrieval and show that the model capacity is not the main bottleneck. It is the optimization failed to fully utilize the network parameters to capture task-specific signals. This motivated our development of TACO-DR, which conducts multi-task learning for universal retrieval with TAsk speCialty Optimization. TACO-DR dynamically adjusts the learning rate for each parameter regrading each task based on its task-specific sensitivity, to encourage parameters to better capture task specific signals. On the KILT benchmark, TACO-DR outperforms various multi-task learning methods and achieves better overall accuracy than single-task models. Our analysis shows that TACO-DR better utilizes the model capacity with more task-specific parameters. Our code and model checkpoints will be open-sourced.

## 1 INTRODUCTION

With pretrained language models (Lee et al., 2019) and dedicated training strategies (Karpukhin et al., 2020a; Xiong et al., 2021), dense retrieval systems now effectively learn a dense representation space that matches queries and relevant documents in nearest neighborhoods. This representation-based retrieval approach provides strong empirical benefits in various scenarios with retrieval as the end goal (Bajaj et al., 2016) and as the first stage retrieval of many language systems (Lewis et al., 2020).

A promising potential of dense retrieval is to unify various scenarios via one representation space, that unifies the representation and match of different types of information, e.g., text and image (Liu et al., 2022), and different types of queries, e.g., keywords, questions, and conversations (Petroni et al., 2021). Such *universal retrieval* (Maillard et al., 2021) leads to instant efficiency benefits, as one document index can support multiple scenarios. It also helps break information boundaries between scenarios with one unified entrance for all user information needs.

Ideally, universal retrieval on multiple scenarios would lead to more accurate retrieval than single scenario systems, with the advantage of multi-task learning. However, recent research observed ambivalent empirical performance of universal retrieval, especially when capturing a large number of retrieval tasks in one universal system (Maillard et al., 2021). This clouds the promise of universal retrieval as its becomes a trade-off between efficiency and accuracy.

In this paper, we conduct thorough investigation on the challenges of multi-task learning in universal retrieval. We performed analysis on the KILT benchmark and found that several state-of-the-art retrieval systems indicate that the network capacity is not yet the main limiting factor for universal retrieval accuracy. Though the multi-task learning has guided the parameters to capture task specific or shared signals, the optimization is not sufficient, resulting in a large fraction of parameters that are not well utilized to capture task-specific signals, as reflected by their low sensitivity (Liang et al., 2022) to each task.

Motivated by our observations, we develop TACO-DR, "TAsk speCific Optimized universal Dense Retriever", that improves universal retrieval by optimizing the task-specialty of neural parameters during multi-task training. TACO-DR first utilizes *task identifier prompts* in its query encoder to

improve the model's task awareness without compromising on the universality of the representation space. Then TACO-DR introduces *task sensitivity guided optimization* (T-SAGE), which dynamically adjusts the gradient step size of each parameter according to its sensitivity to different tasks, to encourage parameters to capture more task specific signals.

To demonstrate the advantages of TACO-DR, we conduct experiments on the eight retrieval tasks included in the standard KILT benchmark, including scenarios such as fact check, entity linking, slot filling, OpenQA, and dialog (Petroni et al., 2021). While standard multi-task dense retrieval failed to outperform their single task counter parts, TACO-DR outperforms both single task models as well as other more advanced multi-task learning techniques. Our ablation confirms the source of effectiveness from its task identifier prompts and task-sensitivity guided optimization.

Our further analysis reveals how TACO-DR better leverages the model parameters for universal retrieval. Compared to standard multi-task learning, TACO-DR activates a larger fraction of the model parameters for each single task. It effectively encourages more model parameters to capture task specific signals, which is achieved by enforcing the parameters to continuously focus on tasks that they are initially learning during optimization. As a result, TACO-DR better utilizes the model parameters to capture the various training signals from multiple tasks and achieve strong retrieval accuracy with one universal retrieval system for many scenarios.

## 2 RELATED WORK

Learning text representations other than discrete bag-of-words has been a long desired goal in information retrieval (Deerwester et al., 1990; Huang et al., 2013). With the benefits of pretrained language models (Kenton & Toutanova, 2019; Lee et al., 2019) and effective hard negative sampling in finetuning (Karpukhin et al., 2020a; Xiong et al., 2021), dense retrieval systems have shown strong effectiveness on many retrieval scenarios (Qu et al., 2020; Herzig et al., 2021; Chang et al., 2021).

Providing a centralized entry to multiple information sources has various advantages. It is more convenient for the user, reduces the information barrier between scenarios, and provides more diverse information access. Previously, this was achieved by complex divide-and-conquer systems, e.g., in federated search (Arguello et al., 2009). Universal retrieval provides a simpler solution with one unified representation space empowering multiple scenarios (Sciavolino, 2021), by embedding all data formats using neural encoders (Baevski et al., 2022).

Recent research on universal retrieval often leverages multi-task learning to train one dual-encoder model for multiple retrieval tasks. Karpukhin et al. (2020a) train a DPR model on four OpenQA tasks and observed accuracy gains on some tasks. Liu et al. (2022) shows the effectiveness of one unified embedding space to retrieve both texts and images. When the number of retrieval tasks grows, multi-task learning becomes challenging. Multi-task trained DPR on eight KILT tasks (Petroni et al., 2021) does not provide much accuracy improvements over single task models (Maillard et al., 2021). Autoregressive retrieval, which directly generates document identifiers using query as input, is another potential solution (De Cao et al., 2020), but currently it mainly thrives on scenarios where the target corpus is small (Bevilacqua et al., 2022) or natural document identifiers exist (e.g., entity names (Chen et al., 2022)).

Recent research has observed various optimization challenges in multi-task learning with modern deep learning systems (Ruder, 2017), for example, gradient conflicts, task imbalance, high variance of gradient magnitudes, to name a few (Yu et al., 2020). Various techniques have been developed. A common way to address these challenges is to perform an operation on the gradients during scholastic updates, e.g., by projecting gradients from different tasks to their norms(Yu et al., 2020), modifying their directions and magnitudes (Wang et al., 2020), and encouraging updates toward common directions (Piratla et al.).

## 3 PRELIMINARIES

In this section we recap preliminaries of dense retrieval and the constraints of universal retrieval.

**Dense Retrieval.** Given a query $q$ and a corpus $C$, the retrieval task is to find a set of relevant documents $d \in C$ for $q$, often by using a retrieval function $f(q, d; \theta)$. Dense retrieval (DR) systems

use a representation-centric solution for retrieval (Lee et al., 2019):

$$f(q, d; \theta) = \text{sim}(g(q; \theta_q), g(d; \theta_d)). \tag{1}$$

It first encodes $q, d$ using the encoder $g(\cdot; \theta_q)$ and $g(\cdot; \theta_d)$, often finetuned from pretrained language models, e.g., BERT (Kenton & Toutanova, 2019) and T5 (Ni et al., 2022). The encoders can share the same parameter $\theta_q = \theta_d$ or use different weights. The similarity metric (sim()) is simple, such as dot product and cosine which support efficient nearest neighbor search (Johnson et al., 2019).

The standard training in dense retrieval is to use NCE loss with sampled negatives (Karpukhin et al., 2020b; Xiong et al., 2021):

$$L(\theta) = -\sum_q \log \left( \frac{\exp(f(q, d^+; \theta))}{\sum_{d' \in \{d^+\} \cup \mathbf{NEG}(q, d^+)} \exp(f(q, d'; \theta))} \right). \tag{2}$$

The sampled negatives $\mathbf{NEG}(q, d^+)$ often consist of random negatives and hard negatives (top retrieved documents from a retrieval model, for example, BM25 (Karpukhin et al., 2020b) or dense retriever $f()$ itself (Xiong et al., 2021).)

**Universal Retrieval** aims to model multiple retrieval tasks $\mathbf{T}$, each with different (but possibly overlapping) queries $q^t$ and corpus $C^t$, using one unified representation space (Maillard et al., 2021).

The key of universality is to map queries and documents into the same embedding space:

$$\forall t \in \mathbf{T} : f(q^t, d^t; \theta) = \text{sim}(g^t(q^t; \theta_q^t), g(d^t; \theta_d)). \tag{3}$$

With different retrieval tasks sharing the same similarity metric in the unified space, universal retrieval provides a *unified information entry* for users to conveniently access different resources, for example, one search place for web, email, enterprise documents, and multi media content. It also enables cross scenario retrieval, with relevance documents from the corpus of a different scenario than the query.

Another advantage of universal retrieval is *efficiency*. Building and maintaining a document index per scenario is costly, and sometimes infeasible with large scale corpora like the web. Unifying the document embedding space enables serving multiple scenarios with one index. To obtain this efficiency benefit, it is often necessary to use the same document encoder $g(\cdot; \theta_d)$ across multiple retrieval tasks, as there is often overlap between task corpora. The query encoder is more flexible (Maillard et al., 2021). Maintaining task-specific query encoders is not as expensive and the query task is often available, e.g., designated by users or predicted by a query classifier (Li et al., 2008).

A natural solution to universal dense retrieval is to train the encoders using multi-task learning:

$$\mathcal{L}(\theta) = \sum_{t \in \mathbf{T}} w^t L^t(\theta) = -\sum_{t \in \mathbf{T}} w^t \sum_{q^t} \log \left( \frac{\exp(f(q^t, d^{t+}; \theta))}{\sum_{d' \in \{d^{t+}\} \cup \mathbf{NEG}(q^t, d^{t+})} \exp(f(q^t, d'; \theta))} \right). \tag{4}$$

It combines the loss from all tasks and balances them by the hyperparameter $\omega^t$.

## 4 ANALYSIS ON MULTI-TASK LEARNING IN UNIVERSAL RETRIEVAL

Ideally, the multi-task learning in Eqn. 4 should provide effectiveness benefits with shared signals between tasks. However, recent research in universal retrieval often observed worse overall retrieval accuracy in comparison with task specific model (Maillard et al., 2021; Kojima et al., 2022). In this section we conduct analysis to understand the challenges of multi-task learning in universal dense retrieval. We first briefly introduce the analysis setup, evaluate various multi-task learning models, and then use two new methods to characterize the multi-task learned model.

**Analysis Setup.** We use the KILT benchmark, which includes eight tasks from five knowledge intensive retrieval scenarios and follow the exact settings from Maillard et al. (2021). The detailed information of KILT benchmark is listed in Appendix A.

The results reported in previous research mainly finetune from BERT and its variants. To gather more observations, we train our own DR models using T5 (Raffel et al., 2019), which has shown better empirical results in search tasks (Ni et al., 2022; Kojima et al., 2022).

| Model | Avg R-Precision |
|---|---|
| Reported in Maillard et al. (2021): | |
| DPR | **67.19** / **42.48** |
| MT-DPR | 64.04 / 36.86 |
| Our Models: | |
| Per Task T5-ANCE | **71.38** / 47.89 |
| QSpec-T5-ANCE | 70.87 / **48.07** |
| MT-T5-ANCE | 70.61 / 46.55 |

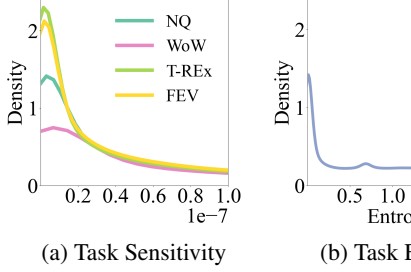

(a) Task Sensitivity   (b) Task Entropy

Table 1: Average page- and passage-level $R$-Precision of single task retrievers versus multi-task universal retrievers on KILT.

Figure 1: Parameter sensitivity analysis of MT-T5-ANCE. The task-specific sensitivity of the other four KILT tasks are in Figure 4. We drop outliers in (a).

Specifically, we use ANCE (Xiong et al., 2021) to finetune the ST5-EncDec architecture (Ni et al., 2022), which takes the text sequences as input to T5's encoder and produces the sequence embedding using the first decoder output on the special token: $g(x, \theta) = \text{T5}(x)$. We experiment with several different parameter sharing variants: Per Task T5-ANCE, the single task model, MT-T5-ANCE, the multi-task model with only one encoder for all, and QSpec-T5-ANCE, which uses task specific query encoders and a unified document encoder.

**Multi-Task Performance.** The performance of these variations are listed in Table 1. All multi-task learned systems underperform single task models. Using a stronger pretrained model, T5, improves performance of all methods but the discrepancy of worse multi-task performance remains. Enriching the capacity of query encoder to task-specific also does not lead to stably better performance than Per Task T5-ANCE. The results indicate that model capacity may not be the bottleneck of universal retrieval. Increasing model capacity, either via a stronger language model or allowing task-specific query encoders, does not lead to better performance than single task models.

**Utilization of Network Parameters.** Next we dive into the learned parameters of our multi-task model and check whether the multi-task training has effectively utilized existing model capacity. For this analysis, we adapt the sensitivity metric, which describes the importance of parameters in a network (Molchanov et al., 2016; 2019), to the multi-task setting. Specifically, we define two sensitivity-based metrics, *task-specific sensitivity* and *task entropy*, to characterize the learned parameters.

*Task-Specific Sensitivity* measures the importance of a parameter to a specific task. The definition of sensitivity is the impact of a parameter w.r.t. a training target based on the changes of loss when the parameter is zeroed out (Molchanov et al., 2019).

Following recent research (Liang et al., 2022), we use the first order Taylor expansion to efficiently calculate task-specific sensitivity:

$$I_t(\theta_i) = |\frac{\partial L^t(\theta)}{\partial \theta_i} \times \theta_i|. \tag{5}$$

It calculates the sensitivity of $\theta_j$ to the loss of task $t$. A higher $I_t(\theta_i)$ means the parameter has captured more signals for the task, as removing it leads to bigger losses on the task. A low sensitivity indicates the parameter is not utilized for the task.

*Task Entropy* describes the spread of a parameter's sensitivity to different tasks:

$$S(\theta_i) = -\sum_t p_t(\theta_i) \log(p_t(\theta_i)); \qquad p_t(\theta_i) = \frac{\exp(I_t(\theta_i)/\tau)}{\sum_{t' \in \mathbf{T}} \exp(I_{t'}(\theta_i)/\tau)}. \tag{6}$$

It first converts the task-specific sensitivity to the multinomial distribution $p_t(\theta_i)$ using softmax and then calculates its entropy. The softmax temperature controls the sharpness of this distribution which we set as 2 to be consistent in all our experiments. For active parameters (with non-zero sensitivity w.r.t. any task), a high task entropy, i.e., a flat distribution $p_t(\theta_i)$, indicates the parameter captures signals shared by multiple tasks. A low entropy reflects a sparse distribution, that the parameter carries signals dedicated to that task (Task-Specific).

---

**Algorithm 1** Task Sensitivity Guided Adaptive Learning (T-SAGE).

---

**Require:** Flatten model parameter $\theta \in \mathbb{R}^M$; minibatches $\mathcal{B}$ where each batch $B \in \mathcal{B}$ is further divided by tasks $B = \{B_t\}_{t \in \mathbf{T}}$; moving average rate $\beta \in [0, 1]$; temperature $\tau > 0$; learning rate $\eta \in \mathbb{R}$

**Notations:** `median` : $\mathbb{R}^{M \times |\mathbf{T}|} \to \mathbb{R}^{|\mathbf{T}|}$ is the column-wise median; `softmax` : $\mathbb{R}^{M \times |\mathbf{T}|} \to \mathbb{R}^{M \times |\mathbf{T}|}$ is the row-wise softmax; $\mathbf{1}_{|\mathbf{T}|}$ is a vector of $|\mathbf{T}|$ ones; $\odot$ is the Hadamard product.

1: Initialize $I \leftarrow \mathbf{0} \in \mathbb{R}^{M \times |\mathbf{T}|}$.
2: **for** each batch $B = \{B_t\}_{t \in \mathbf{T}}$ in $\mathcal{B}$ **do**
3:     Compute the task-specific loss $L_t(\theta)$ on $B_t$ for each $t \in \mathbf{T}$.
4:     Compute the gradient matrix $G \in \mathbb{R}^{M \times |\mathbf{T}|}$ with each column $G_t \leftarrow \nabla L_t(\theta)$.
5:     Compute the sensitivity matrix $I' \in \mathbb{R}^{M \times |\mathbf{T}|}$ with each column $I'_t \leftarrow G_t \odot \theta$.
6:     Normalize the sensitivity scales across tasks $I' \leftarrow I' \mathrm{diag}\left(\texttt{median}(I')\right)^{-1}$.
7:     Update the moving average $I \leftarrow \beta I + (1 - \beta)I'$.
8:     Update the parameter $\theta \leftarrow \theta - \eta(G \odot U)\mathbf{1}_{|\mathbf{T}|}$ where $U \leftarrow \texttt{softmax}(I/\tau) \in \mathbb{R}^{M \times |\mathbf{T}|}$.
9: **end for**

---

We leverage these two tools to study the learned parameter of MT-ANCE-T5, our multi-task learned model. The kernel density estimated distribution of the learned parameters are in Figure 1. From the plot we make the following two observations of the multi-task learned model:

1. *The Bad. The training left a significant amount of model capacity on the table.* Figure 1a shows a long tail distribution of task-specific sensitivity, with majority of parameters under utilized for each task.

2. *The Good. As desired, the multi-task learning pushes some parameters to capture task specific signals and some to capture shared signals.* Figure 1b reveals the bimodal distribution with two peaks towards each end of the entropy.

These observations indicate that a major bottleneck in universal retrieval is that the optimization process does not fully utilize the network parameters. Our analysis also shows an encouraging sign that the parameters have the tendency to capture task-specific signals in multi-task learning, only not as effective as in single-task models.

## 5 TASK-SPECIALTY OPTIMIZATION

In this section, we propose TACO-DR, which aims to unleash the model capacity for universal retrieval using TAsk speCific Optimization. TACO-DR first uses prompt to convey the task identifier in the query. Then, motivated by our analysis in the last section, it employs a task sensitivity guided adaptive learning (T-SAGE) that dynamically adjusts learning rates to encourage task specialty on parameters in multi-task learning.

**Task Identification Prompts.** TACO-DR uses the task name prompts (Raffel et al., 2019) in front of the query to mark the task for the universal retrieval model:

$$q' = \text{Task Name} \circ \texttt{[SP]} \circ q. \tag{7}$$

It concatenates ($\circ$) the task name string to the search query with a special token (`[SP]`). This is a standard way in prompt-based learning to notify the model of the origin of the query. As we discussed in Section 3, the query side in universal retrieval is more flexible and can have different query formats for different tasks. The task prompts can be customized by the user ("prompt engineering") or being predicted by query classifiers if needed. All the remaining parts of the model are kept the same as MT-T5-ANCE: T5 encoder shared for both query and document, dot product similarity, standard document inputs, and the same encoder model for all tasks.

**Task Sensitivity Guided Adaptive Learning (T-SAGE).** To encourage model parameters to capture task specific signals, TACO-DR uses the task sensitivity distribution to dynamically change the learning rate of each parameter during optimization. Recent research has shown the effectiveness of sensitivity guided adaptive learning in improving network parameter utilization (Liang et al.,

2022). T-SAGE extends it to multi-task learning by assigning an adaptive learning rate for each parameter-task pair.

Formally, in the stochastic learning process, T-SAGE performs the $k$-th gradient step on $\theta_i$ as:

$$\theta_i^{k+1} = \theta_i^k - \eta \sum_{t \in \mathbf{T}} u_t^k(\theta_i) \frac{\partial L^t(\theta^k)}{\partial \theta_i^k}, \tag{8}$$

where $\eta$ is the overall gradient step size, $\partial L^t(\theta^k)/\partial \theta_i^k$ is the gradient of task $t$ on parameter $\theta_i$, and $u_t^k(\theta_i)$ is a task specific learning rate for each parameter.[1]

As shown in Figure 1b, in the multi-task learning, network parameters have the tendency to capture task specific signals or shared signals. Ideally, T-SAGE uses $u_j^t$ to encourage this tendency, for parameters who have a preference towards one task to commit on learning task specialty, and the rest to keep learning shared signals.

For that goal, we first calculate the task-specific sensitivity for parameters:

$$I_t^k(\theta_i^k) = |\frac{\partial L^t(\theta^k)}{\partial \theta_i^k} \times \theta_i^k|. \tag{9}$$

$$\bar{I}_t^k(\theta_i) = I_t^k(\theta_i)/\text{Median}_{1 \leq m \leq M}(I_t^k(\theta_m)); \tag{10}$$

$$\hat{I}_t^k(\theta_i) = \beta \bar{I}_t^k(\theta_i) + (1 - \beta)\hat{I}_t^{k-1}(\theta_i). \tag{11}$$

The per-step task-specific sensitivity $I_t^k(\theta_i^k)$ has a high variance in stochastic training. Eqn. 10 normalizes sensitivity of a parameter using the median of all parameters w.r.t. to this task. It mitigates the variances between different tasks. We use median instead of mean to account for the long tail distribution of task-specific sensitivity. Eqn. 11 introduces a momentum mechanism to improve the estimation. It avoids sudden changes of task sensitivity based on one mini-batch.

Then we use the parameter's task specialty distribution to define its dynamic learning rate:

$$u_t^k(\theta_i) = \frac{\exp(\hat{I}_t^k(\theta_i)/\tau)}{\sum_{t' \in \mathbf{T}} \exp(\hat{I}_{t'}^k(\theta_i)/\tau)}. \tag{12}$$

Eqn. 12 dynamically sets the learning rate of parameter $\theta_i$ for task $t$ by comparing its task sensitivity $\hat{I}_t^k(\theta_i)$ to other tasks. If $\theta_i$ is more sensitive to a task $t$, it receives a larger rate $u_t^k(\theta_i)$ to capture more task specific signals. If the parameter has a flat distribution, it receives balanced learning rate over multiple tasks, with the potential to learn shared signals between multiple tasks.

The emphasis on task specialty is controlled by the softmax temperature $\tau$. A smaller $\tau$ produces a sharper distribution and emphasizes more on learning task specialty. We can first employ a "burn-in" period that trains with uniform learning rates or high temperature $\tau$, for parameters to declare their tendency. Then we can reduce the temperature to encourage the learning of task specialty and to better utilize the model parameters. The matrix version of T-SAGE is summarized in Algorithm 1 for implementation references.

## 6 EXPERIMENTAL SETUP

We briefly describe our experimental settings in this section. More details can be found in Appendix C.

**Datasets.** We follow Maillard et al. (2021) and use eight tasks from KILT (Petroni et al., 2021). We randomly down sample the training data of the two largest datasets (T-REx and zsRE) to the same order of magnitude as the rest. All the datasets share the same document corpus of 36 million disjoint 100-token Wikipedia passages preprocessed by Maillard et al. (2021).

**Evaluation.** We report page-level $R$-precision and passage-level $R$-precision. We use official KILT evaluation scripts to evaluate page-level $R$-precision and TREC Eval[2] for passage level.

---

[1] We derive in standard SGD. Changing to momentum based method is straightforward.

[2] https://trec.nist.gov/trec_eval/

| Model | Fact Check.
FEV | Ent. L.
AY2 | Slot Filling | | Open Domain QA | | | Dial.
WoW | Avg |
|---|---|---|---|---|---|---|---|---|---|
| | | | T-REx | zsRE | NQ | HoPo | TQA | | |
| BM25 | 50.13 / 40.06 | 3.47 | 58.60 / 51.64 | 66.43 / 52.98 | 25.83 / 14.20 | 43.95 / 38.38 | 29.44 / 16.16 | 27.50 / 18.41 | 38.17 / 33.12 |
| **Autoregressive Models with Large-Sized Models (For Reference).** | | | | | | | | | |
| MT-BART† | 81.92 / - | 89.17 | 75.18 / - | 91.08 / - | 58.62 / - | 48.69 / - | 67.64 /- | 50.98 / - | 70.41 / - |
| MT-CorpusBrain† | 82.06 / - | 90.84 | 77.62 / - | 98.26 / - | 59.10 / - | 50.07 / - | 68.78 / - | 53.75 / - | 72.56 / - |
| **Dual Encoder Dense Retrieval Models with Base-sized Models.** | | | | | | | | | |
| Per Task DPR V2* | 73.60 / 43.92 | 81.77 | 69.08 / 58.54 | 97.74 / 78.81 | 63.24 / 28.13 | 46.63 / 43.47 | 65.12 / 23.79 | 40.32 / 20.73 | 67.19 / 42.48 |
| MT-DPR V2* | 74.72 / 46.96 | 83.78 | 69.18 / 53.54 | 77.23 / 41.70 | 61.51 / 28.80 | 44.21 / 38.42 | 61.95 / 24.56 | 39.70 / 24.07 | 64.04 / 36.86 |
| Per Task ANCE | 74.28 / 44.89 | 85.28 | 77.18 / 72.09 | **99.38 / 84.47** | **65.39 / 33.14** | 46.79 / 43.40 | 69.08 / **29.57** | 53.63 / 27.64 | 71.38 / 47.89 |
| QSpec-T5-ANCE | 82.71 / 57.91 | **87.56** | 72.72 / 66.90 | 85.15 / 71.37 | 64.01 / 32.92 | 49.74 / 45.80 | 69.12 / 29.18 | 55.93 / 32.38 | 70.87 / 48.07 |
| MT-T5-ANCE | 84.03 / 57.57 | 85.62 | 70.96 / 65.15 | 86.04 / 69.79 | 62.46 / 30.51 | 49.78 / 44.35 | 66.04 / 25.93 | 59.95 / 32.55 | 70.61 / 46.55 |
| PCG-T5-ANCE | 84.65 / 58.60 | 85.51 | 70.48 / 65.01 | 85.02 / 70.65 | 62.14 / 31.22 | **50.93** / 46.26 | 67.57 / 27.97 | 59.82 / 32.68 | 70.77 / 47.48 |
| CGD-T5-ANCE | 83.15 / 54.95 | 80.96 | 66.54 / 60.32 | 79.03 / 59.49 | 62.32 / 29.04 | 48.71 / 43.26 | 65.18 / 27.47 | 58.87 / 32.81 | 68.10 / 43.91 |
| GN-T5-ANCE | 84.10 / 57.39 | 85.18 | 70.08 / 64.45 | 84.94 / 68.61 | 62.71 / 29.43 | 49.09 / 43.24 | 64.43 / 25.09 | 60.05 / 32.68 | 70.07 / 45.84 |
| TACO-DR | **86.17 / 60.76** | 84.64 | **78.12 / 72.57** | 97.91 / 82.80 | 61.86 / 31.16 | 50.61 / **46.72** | **69.62** / 28.32 | **60.97 / 33.24** | **73.74 / 50.80** |

Table 2: Page/passage level R-precision on KILT validation data. **Bold** indicates the best dual encoder model and underline indicates the second. Only page-level retrieval is defined for AIDA-YAGO 2 (AY2). † and ∗ mark results from Chen et al. (2022) and Maillard et al. (2021).

**Model configuration.** We use the standard dual-encoder architecture, initialized with T5-base (Raffel et al., 2019) and take the first decoder output as the sentence embedding (Ni et al., 2022). The query encoder and passage encoder share weights in all our runs except QSpec-T5-ANCE.

**Training details.** We train MT-T5-ANCE from T5-base using vanilla multi-task learning, for 20 epochs using BM25 negatives (Karpukhin et al., 2020b) and then seven ANCE (Xiong et al., 2021) episodes with its own hard negatives refreshed at the beginning of each episode. We use a similar process for TACO-DR but with uniform weights (infinite softmax temperature) in T-SAGE for the first six ANCE episodes and $\tau = 2$ in the last episode. Training TACO-DR with an annealed temperature earlier in the process achieves similar results. We provide more details in Appendix C. We use Adam (Kingma & Ba, 2015) with a linear learning rate decay schedule with warmup proportion 0.1 over 3 epochs for each ANCE iteration.

**Baselines.** We compare our model with recent universal retrieval models (Maillard et al., 2021; Chen et al., 2022). We also compare with general multi-task learning models PCG (Yu et al., 2020), CGD (Piratla et al., 2022) and GN (Chen et al., 2018) (using our own implementations). A parallel work uses autoregressive models to generate entity names (as document identifier) to conduct retrieval on KILT (Chen et al., 2022). They use large size models and a large amount of additional training data. We list their models trained solely on KILT training labels for reference.

## 7 EVALUATION RESULTS

This section presents four experiments on TACO-DR's retrieval accuracy, its ablations, learned task specialty, and its parameter commitment on tasks.

### 7.1 OVERALL RETRIEVAL ACCURACY

Table 2 lists the retrieval accuracy of evaluated methods. We evaluate R-precision at both page and passage level but will mainly focus on page level evaluation in later experiments, which is the metric designed for many tasks (Kwiatkowski et al., 2019; Nie et al., 2019, e.g.).

TACO-DR outperforms all other methods on the average accuracy. It achieves best single task page-level R-precision on four of the eight KILT tasks, the most robust among all evaluated methods. TACO-DR is the only evaluated universal retrieval method that outperforms its task-specific counterpart, Per Task T5-ANCE, on both page and passage level average accuracy. Though the autoregressive models employ a large sized pretrained model and benefit from the existence of semantic document identifiers (entity name) in KILT, TACO-DR achieves better overall accuracy with a smaller model when all are trained on KILT data.

Recent multi-task learning techniques, PCG (Yu et al., 2020), CGD (Piratla et al., 2022) and GN (Chen et al., 2018), do not outperform standard multi-task learning in universal retrieval. These methods mainly focus on mitigating the conflicts of gradients from different tasks. Our analysis shows the

| Variants | Fact Check. FEV | Ent. L. AY2 | Slot Filling T-REx | zsRE | Open Domain QA NQ | HoPo | TQA | Dial. WoW | Avg |
|---|---|---|---|---|---|---|---|---|---|
| MT-T5-ANCE | 84.03 | 85.62 | 70.96 | 86.04 | 62.46 | 49.78 | 66.04 | 59.95 | 70.61 |
| QSpec-T5-ANCE | 82.71 | **87.56** | 72.72 | 85.15 | **64.01** | 49.74 | 69.12 | 55.93 | 70.87 |
| MT-T5-ANCE w. Marker | 84.49 | 85.51 | 73.88 | 89.37 | 62.85 | 50.97 | 67.70 | 60.02 | 71.85 |
| TACO-DR w.o. T-SAGE | 84.81 | 85.49 | 75.00 | 92.24 | 62.81 | **51.47** | 68.95 | 60.54 | 72.66 |
| TACO-DR w.o. Prompt | 85.71 | 84.68 | 74.82 | 94.68 | 61.05 | 49.38 | 67.79 | 58.81 | 72.12 |
| TACO-DR | **86.17** | 84.64 | **78.12** | **97.91** | 61.86 | 50.61 | **69.62** | **60.97** | **73.74** |

Table 3: Page level accuracy of TACO-DR variants. MT-T5-ANCE w. Marker uses the coarse task marker from Maillard et al. (2021). Best results are **bolded**.

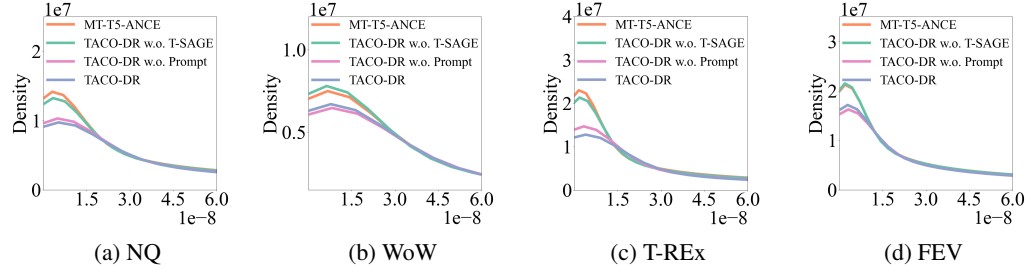

| (a) NQ | (b) WoW | (c) T-REx | (d) FEV |

Figure 2: Task-specific sensitivity distribution on the training data of four KILT tasks. The final models are used. The distribution of the other four tasks are shown in Figure 4 in Appendix. The x-axis is sensitivity, and we drop outliers that are far from the median to ease visualization.

challenge is mainly on under captured task specific signals and TACO-DR is designed to increase the utilization and task specialty of model parameters. Section 7.3 further analyzes this advantage of TACO-DR.

Note that no method performs the best on every retrieval task. Universal retrieval is challenging with scenarios consisting of variable training data quantity, task difficulty, and different user preferences. How to achieve robust effectiveness benefits is a next step of development for universal retrieval.

## 7.2 ABLATION STUDIES

The page level results of TACO-DR variants and corresponding baselines are shown Table 3.

Using Task Identification Prompt in standard multi-task learning also improves its performance. TACO-DR w.o. T-SAGE outperforms MT-T5-ANCE w. Marker where the only difference is the task prompts. Both outperform QSpec-T5-ANCE which uses significantly more parameters, further confirming that the model capacity is not the biggest bottleneck in universal retrieval.

Both task prompt and T-SAGE contribute to the effectiveness of TACO-DR. The two target different parts of multi-task learning. The task prompt conveys the task specific information in model inputs and T-SAGE adjusts the learning process to emphasize task specialty during optimization. The benefits of prompt are more well understood in multi-task learning (Raffel et al., 2019) and in the next experiments we focus on the properties of T-SAGE.

## 7.3 PARAMETER UTILIZATION AND TASK SPECIALTY

In this experiment we analyze TACO-DR's parameters using *task-specific sensitivity* and *task entropy*.

Figure 2 plots the kernel density estimated distribution of task-specific sensitivity in TACO-DR and its variants. There is a significant reduction of non-activated parameters in TACO-DR compared with MT-T5-ANCE. This mainly attributes to T-SAGE. Using it, with or without prompt, significantly reduces the peak on the low sensitivity side, showing a better utilization of parameter capacity for each task (Liang et al., 2022).

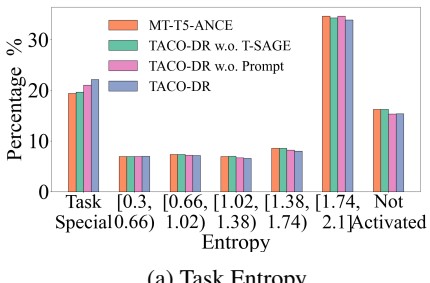
(a) Task Entropy.

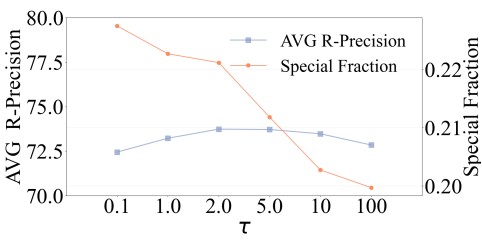
(b) Influence of T-SAGE Temperature.

Figure 3: The analysis of task specialty on parameters learned by different methods (a) and when using different temperatures in T-SAGE.

Figure 3a plots the histograms of task entropy for the learned parameters. We first group parameters into two special bins. The first is a "Task Specific" bin that includes parameters whose entropy is smaller than 0.3, which is the entropy of 95% probability on one task and the 5% uniformly on the rest seven. The "Not Activated" bin includes parameters whose sensitivity w.r.t. all tasks is near zero ($< 0.1$). TACO-DR significantly improves the fraction of task specific parameters to 22%, in comparison with 19% in MT-T5-ANCE. It also reduces the fraction of not activated parameters, showing optimizing task specialty also better utilizes the model capacity.

Figure 3b illustrates the influence of the softmax temperature $\tau$ in Eqn. 12, which controls the sharpness of the sensitivity distribution thus the encouragement of task specialty. The accuracy of TACO-DR is quite stable w.r.t. different $\tau$ values in normal temperature range. At the same time, it does show the effectiveness of $\tau$ in controlling for different levels of task specialty. A smaller $\tau$ leads to more task specific parameters and a bigger $\tau$ leads to fewer ones.

### 7.4 COMMITMENT OF TASK SPECIFIC PARAMETERS DURING OPTIMIZATION

This last experiment studies the momentum mechanism in T-SAGE. Besides retrieval accuracy, we also calculate the task commitment rate, which is a fraction of task-specific parameters that are committed to the same task after some training steps (400), i.e., being most sensitive to the same task. The results are in Table 4.

At the early stage of optimization, the task commitment rates are low. Parameters are still wandering around signals from different tasks. Strong

| $\beta$ | 0 | 0.6 | 0.7 | 0.8 | 0.9 | 0.999 |
|---|---|---|---|---|---|---|
| **Avg R-prec** | 72.91 | 72.98 | 73.02 | 73.16 | 73.61 | 73.74 |
| **Step** | **Task Commitment Rate** | | | | | |
| 400 | 43.8% | 41.2% | 41.4% | 39.7% | 36.5% | 35.5% |
| 800 | 61.6% | 57.8% | 58.8% | 56.4% | 56.5% | 52.0% |
| 1200 | 63.4% | 58.5% | 60.0% | 59.5% | 61.1% | 56.4% |
| 1600 | 68.7% | 65.4% | 68.3% | 67.8% | 70.4% | 70.2% |
| 2000 | 72.5% | 74.3% | 73.8% | 72.5% | 75.9% | 76.3% |

Table 4: Impact of momentum decay factor $\beta$. The commitment rate is calculated by comparing the task distribution at the corresponding step to the previous row.

momentum factor carries more random signals from early chaotic stages and in fact hurts the commitment rate. With enough training, parameters start to settle to their target tasks and we observe significant increments of commitment rates, even for the ones without momentum $\beta = 0$. There a stronger momentum provides better commitments and eventually leads to better retrieval accuracy.

## 8 CONCLUSION

In this paper we present TACO-DR that unleashes the model capacity for universal retrieval by encouraging parameter task specialties in multi-task training. We first analyze the standard multi-task learning in universal retrieval and show that much of the model capacity is under utilized. TACO-DR dynamically adjusts the learning rate of each parameter for each task based on its task sensitivity, to encourage the learning of task specific signals during stochastic training. Our experiments demonstrate the benefits of TACO-DR, its optimization of task specialty, better utilization of network parameters, and effective control of the optimization process.

## REPRODUCIBILITY STATEMENT

To enhance reproducibility, we describe our model configuration and overall training details in Section 6. Beyond of that, we include more details and hyperparameters in Appendix. Appendix A shows data statistics and data-related hyperparameters. Appendix C provides batch sampling strategy, training hyperparameters and more training details. We plan to submit our code after the discussion forums open. We will make a comment containing a link of our anonymous repository internally visible to reviewers and ACs. We will release our code and model checkpoints, along with analysis scripts if this work is accepted.

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

## A    DATA STATISTICS

See Table 5 for data statistics and some data-related hyperparameters. We randomly downsample T-REx and zsRE to bring them to the same order of magnitude as the others.

| Dataset | #Train | Batch size | Max query length |
|---|---|---|---|
| Natural Questions | 76445 | 16 | 32 |
| TriviaQA | 52886 | 14 | 32 |
| HotpotQA | 68659 | 15 | 36 |
| Wizard of Wikipedia | 79535 | 16 | 256 |
| T-REx | 94514 | 16 | 32 |
| FEVER | 70757 | 15 | 64 |
| Zero Shot RE | 99500 | 17 | 32 |
| AIDA-YAGO 2 | 17895 | 11 | 128 |

Table 5: Data statistics and some data-related hyperparameters for our experiments. The original T-REx and zsRE have very large dataset size. We randomly downsample T-REx and zsRE to bring them to the same order of magnitude as the others.

| | Fact Check. | Ent. L. | Slot Filling | | Open Domain QA | | | Dial. | |
|---|---|---|---|---|---|---|---|---|---|
| Model | FEV | AY2 | T-REx | zsRE | NQ | HoPo | TQA | WoW | Avg |
| Annealed TACO-DR | 86.10 / 59.97 | 84.26 | 77.56 / 72.15 | 97.15 / 82.28 | 61.30 / 30.45 | 50.60 / 46.69 | 69.34/ 27.91 | 61.43 / 33.69 | 73.47 / 50.45 |
| TACO-DR | 86.17 / 60.76 | 84.64 | 78.12 / 72.57 | 97.91 / 82.80 | 61.86 / 31.16 | 50.61 / 46.72 | 69.62 / 28.32 | 60.97 / 33.24 | 73.74 / 50.80 |

Table 6: Page- and passage- level R-precision for TACO-DR with annealed temperature started from 4-th ANCE episode with annealing schedule $\tau = 64 \times (0.5)^{a-4}$ where $a >= 4$ is the ANCE episode number. We also include the performance of TACO-DR trained with the setting in section 6 for reference.

| learning rate | warmup ratio | # negatives per $q$ | epochs | $\tau$ | $\beta$ | total batch size |
|---|---|---|---|---|---|---|
| 5e-6 | 0.1 | 2 | 3 | 2.0 | 0.999 | 120 |

Table 7: Training hyperparameters for training our TACO-DR model. We use AdamKingma & Ba (2015) with learning rate $5e - 6$. We use linear learning rate schedule with warmup raio 0.1. Each query uses 2 hard negatives for training. Each ANCE episode trains for 3 epochs. Total batch size of all task batches are 120.

## B  OTHER PLOTS

Figure 4 shows parameters sensitivity distribution of the remaining 4 tasks TQA, HoPo, zsRE and AY2 that Figure 2 doesn't show. We observe similar trend to Figure 2 that T-SAGE significantly reduce non-activated parameters except for zsRE. We note that zsRE is a very simple task that we can achieve almost $100\%$ accuracy, which doesn't require many activated parameters.

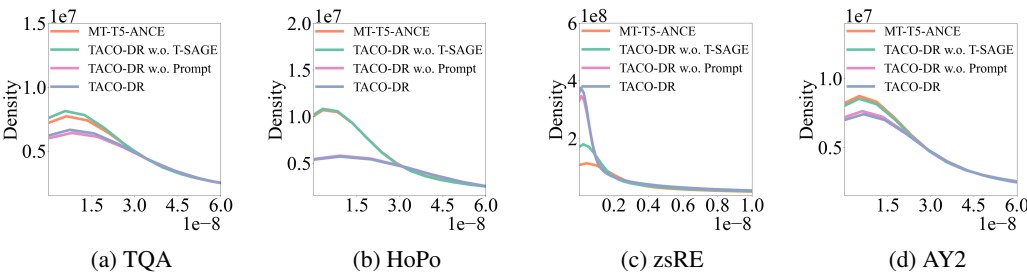

(a) TQA          (b) HoPo          (c) zsRE          (d) AY2

Figure 4: Parameters sensitivity distribution using kernel density estimation on the training data of the remaining four KILT tasks that Figure 2 doesn't show.

## C  MORE EXPERIMENT DETAILS

**Batch Sampling**  We follow Raffel et al. (2019) and use temperature-scaled mixing sampling strategy to compute batch size for each task $t \in \mathbf{T}$: $B_t \propto (N_t / \sum_{t' \in \mathbf{T}} N_{t'})^{1/c}$ for some temperature $c$ (we set it to 4 in our experiments). Here $N_t$ is the dataset size of task $t$. Note that we compute task loss of each task batch independently instead of mixing all task batches for every optimization step. Each dataset needs to sample different number of batches to cover every training sample in that dataset once. We set the maximum of them as the number of batches that every dataset needs to sample. We shuffle and cycle batch sampling iterators of datasets that finish iterating early. Batch size of each dataset computed by setting mixing temperature $c = 4$ and $\sum_{t' \in \mathbf{T}} N_{t'} = 120$ is in Table 5

**Other training details**  The data-related hyperparameters, such as maximum input query length and batch size, are listed in Table 5. The training hyperparameters are listed in Table 7. We use NCE loss with cross device in-batch negative mixed with hard negatives to compute each task loss. We sample two hard negatives for each query. We employ a "burn in" period for the first $10\%$ training steps with uniform learning rates for parameters to declare their tendency during T-SAGE optimization. All of our experiments are run on a machine with 8 A100-80GB GPUS. Our implementations are built upon OpenMatch (Liu et al., 2021).

**Annealed Softmax Temperature** We also try annealing the softmax temperature in T-SAGE early at the beginning of the fourth ANCE episode. Specifically, we anneal the softmax temperature according to schedule $\tau_a = 64 \times (0.5)^{a-4}$ where $a >= 4$ is the $a$-th ANCE episode. The results are listed in Table 6 We achieve similar performance as the setting in section 6. The training setting in section 6 has low cost with slightly higher performance compared to this annealed setting. .

