# OpenReview forum: "Unleash Model Capacity for Universal Dense Retrieval by Task Specialty Optimization"
_ICLR.cc/2023/Conference — Submitted to ICLR 2023_

### Official Review · Reviewer_6W5P · 2022-10-23

**Confidence:** 4
**Correctness:** 3
**Technical Novelty And Significance:** 2
**Empirical Novelty And Significance:** 2
**Recommendation:** 5

**Clarity, Quality, Novelty And Reproducibility:**

The paper is generally clearly written.

The major concern is novelty. First, the paper mainly applies an existing adaptive method to the universal retrieval task, which is one kind of multi-task learning in general. It is good to show that the phenomenon for retrieval is mainly due to “under captured task specific signals”, but it is one kind of problem in general multi-task learning.

Though the proposed method is shown to perform better than few multi-task learning optimization methods that focus on gradient conflicts, there is a rich literature of multitask learning that deals with other concerns, and the paper does not discuss why the proposed method works better (again, the optimization used in this paper is an application, not really a brand new algorithm). For example, the classical GradNorm (GradNorm: Gradient Normalization for Adaptive Loss Balancing in Deep Multitask Networks. ICML2018) paper is not cited or discussed, which shares similar spirits by adjusting gradient magnitudes/learning rates. There are many follow-ups of GradNorm.

The other major technique section, the prompt trick, is commonly used in the literature so they do not contribute much to novelty.

Reproducibility should be fair but not very clear. The authors claim “code and model checkpoints will be open-sourced” but not yet.


**Details Of Ethics Concerns:**

One universal retriever across different corpuses may have privacy concerns.

**Strength And Weaknesses:**

Strength
The paper is generally well written, though some details can be provided more properly.
The studied problem is relatively new.

Weakness
Novelty. The analysis and method used in this paper are applications of existing methods. Though it is a good effort, this is unlikely to be sufficient for a top-tier ML venue.
Given the rich literature of multi-task learning, there are methods that do not focus on gradient conflicts. The chosen related baselines are likely to be used in the wrong context.


**Summary Of The Paper:**

The submission concerns universal retrieval, or multi-task dense retrieval, where queries and documents from different sources are mapped to the same space. The paper first conducts a study using T5 on the KILT dataset and empirically shows that there is some gap between single task performance vs multi-task. Then parameter importance analysis is done using existing techniques, showing that it is mainly due to parameters under capturing task specific signals. Then an optimization with adaptive learning rates is applied. Experiments on the KILT shows the proposed framework can achieve better average performance than alternatives, including BM25, naive multi-task learning, and two multi-task methods can focus on mitigating gradient conflicts.

**Summary Of The Review:**

Overall the paper deals with a relatively new problem and is clearly written. The novelty is a major concern for a top-tier ML venue (an NLP venue may make more sense). Please see the detailed summary and discussions above.
Some other comments:

“The results indicate that model capacity may not be the bottleneck of universal retrieval” - what size of T5 is used (this should be clarified anyway - looks like BASE is used from experiments)? It looks like the gap is smaller in Table 1. What if larger T5 models are used?

---

> ### Author Response · Authors · 2022-11-18
> **Author Response**
>
> We would like to thank the reviewer for the feedbacks and comments.
>
> - We are not applying existing methods. We are the first to define task sensitivity by parameter sensitivity and use it for multi-task learning. We are not just applying Liang et al. to universal retrieval. Their method is a parameter-level update for unused parameters in a single task. Our method is a task-level update for specializing parameters for multiple tasks. These are completely different problems.
>
> - Our adaptive update is parameter-level as well as task-level. We adjust learning rates along both task dimension and parameter dimension. This is critically different from existing works on gradient adjustment for multitasking which only consider loss-level gradient re-weighting (GradNorm, CGD). To our knowledge, the use of parameter sensitivity—which has largely been used in the context of model pruning only—for multitasking is itself novel.
>
> - We thank the reviewer for the pointer to GradNorm. We have conducted additional experiments with GradNorm and included the results in the general response, copied here for convenience:
>
>
>
> |                  | FEV | AY2 | T-REx | zsRE | NQ | HoPo |  TQA |  WoW | AVG|
> |:------:|:------:|:----------:|:-----------:|:---------:|:---------------------:|:---------------------:|:---------------------:|:---------------------:|:---------------------:|
> | no prompt |  84.1/57.39 | 85.18 | 70.08/64.45 | 84.94/68.61 |  62.71/29.43   |  49.09/43.24 |  64.43/25.09  |   60.05/32.68  |  70.07/45.84  |
> | prompt |  84.7/58.3 | 85.28 | 75.32/69.6 | 91.73/75.54 | 63.8/32.45  |  51.97/47.7 |  69.3/29  |  60.31/34.18  |  72.80/49.54  |
> | TACO-DR| 86.17/60.76| 84.64 | 78.12/72.57| 97.91/82.80| 61.86/31.16| 50.61/46.72 | 69.62/28.32 | 60.97/33.24 | 73.74/50.80|
>
>
> - We see that GradNorm performs generally worse than our method, especially without the added of our prompt technique (70.07 vs 72.12). This highlights the benefit of the fine-grained parameter-level update in our method.
>
> - We think it is very important to highlight the benefit of task identification prompts because they were shown to be not helpful in Maillard et al. We debunk this previous finding by showing that prompts yield clear improvements when the model is properly optimized.
>
> - We use T5-base in all our experiments (it is clearly stated in the submission). Our finding is that a single model is able to outperform much bigger models with task-specific query/document encoders (i.e., bigger model size is not necessarily better performance). We will clarify this point in the final version.

---

### Official Review · Reviewer_UpYV · 2022-10-24

**Confidence:** 3
**Correctness:** 4
**Technical Novelty And Significance:** 3
**Empirical Novelty And Significance:** 3
**Recommendation:** 5

**Clarity, Quality, Novelty And Reproducibility:**

The figures can be improved by adding labels to axes and adding additional explanation about what the figures are showing.

I would also encourage the authors to release their code to facilitate reproducibility.

**Strength And Weaknesses:**

Strengths:
* The authors extend sensitivity guided adaptive learning (Liang et al., 2022) to the multi-task setting.
* Empirically, the authors show that they are able to outperform multi-task and single-task baselines on the KILT benchmark.
* The authors also show that adding task labels as a prompt improves multi-task retrieval performance.

Weakness:
* The sensitivity analysis graphs in  figure 1 and figure 2 are unclear and need more explanations. The x-axis in unlabelled in the graph and it is unclear what the axes are and what range of values they show. In figure 2, is there a tail that is not shown. How come the peak on the left is lower for the TACO methods but the tail is the same for all the methods?
* The percentage ranges for the values in figure 3a and table 4 don't seem to be consistent with each other. Is the commitment rate calculated differently for these?
* It's not clear whether the current framework can handle the case when a parameter is important for a subset of the tasks but not all the tasks.
* This method doesn't seem specific to retrieval. Can other multitask settings also benefit from using task-specific parameter-specific learning rates?

**Summary Of The Paper:**

The authors introduce a multi-task learning method for universal retrieval. They identify that the reason existing approaches fail is because they fail to capture task-specific signals and therefore propose a method that learns task-specific parameter-specific learning rates to better capture task-specific signals.

**Summary Of The Review:**

The presented work does achieve state-of-the-art results for multitask retrieval. TACO-DR is an extension of an existing method (Liang et al., 2022). Although it works well for retrieval, it's not clear whether this method is useful just for this one setting or is more broadly useful. It would be interesting to investigate whether this method provides benefits for other multi-task learning applications.

---

> ### Author Response · Authors · 2022-11-18
> **Author Response**
>
> We would like to thank the reviewer for the comments.
>
> -  Fig 1 a : x-axis label is sensitivity.  Fig 2: since sensitivity distribution has very long tail, we cut off the tail(outliers) by dropping those points/outliers that are too far from the median.
> - Task commitment is a fraction of task-specific parameters that are committed to the same
> task during K training steps, i.e., being most sensitive to the same task. It can reflect the usefulness of momentum. Table 4 has no relation with fig 3(a), we just want to measure the momentum influence in table 4.
> -  Our framework can handle the case when a parameter is important for a subset of the tasks but not all the tasks. We define task sensitivity by tasking softmax over normalized per-task parameter sensitivity along task dimension. If a parameter is only sensitive to a subset of tasks, the model will assign high weights for the task gradients of those sensitive tasks and assign very small weights for non-sensitive tasks gradients.
> -  Thanks for highlighting the possibility of using our technique for other multi-tasking problems. Our technique is indeed not specific to retrieval and can be used for general multi-task learning. We will explore applying it to other problems in future work.

---

### Official Review · Reviewer_KDMG · 2022-10-25

**Confidence:** 3
**Correctness:** 3
**Technical Novelty And Significance:** 2
**Empirical Novelty And Significance:** 2
**Recommendation:** 5

**Clarity, Quality, Novelty And Reproducibility:**

The paper is generally well-written. Related work could better describe the key differences between this paper and existing literature.


What is "special fraction" in Fig. 3 (b)?

Fig. 3 (a) needs to increase such that the x-axis ticks are more readable. Also, it seems that the ablation study shows there is not much difference between the proposed models and MT-T5-ANCE.

Please further explain the task commitment in Table 4, in particular how the task commitment rate is computed. Wouldn't make sense to computer this task commitment rate for the best-performing baselines and present a comparison among models?

**Strength And Weaknesses:**

[+] The analysis of task sensitivity provides a clearly articulated motivation for the proposed work.

[-] In many cases, the performance difference between TACO-DR and the next best baseline is marginal.

[-] The proposed work seems to be a straightforward application of prompt tuning and task sensitivity adaptive learning rate (similar to momentum-based optimizers in the literature). If there exist novel unique contributions, these could be better highlighted in the introduction and related work.

[-] More relevant baselines are needed, e.g., other formulations of task-specific learning rates, e.g., a self-paced version that the task-specific hyper-parameter is weighted based on a loss-based task difficulty score, or an entropy-based task-specific learning rate. If baselines are trained with vanilla gradient descent, it would be interesting to see if standard Adagrad produces the same results.

**Summary Of The Paper:**

This work proposess a multi-task universal retrieval model that is based on task-specific prompt tuning and adaptive learning are based on task sensitivity. Experiments on the KILT benchmark dataset over existing models, and ablation on the task sensitivity, show that although the model capacity of existing works is not sufficiently utilized, universal retrieval models can capture task-specific signals. The proposed method tries to address the limitation of model capacity utilization.

**Summary Of The Review:**

The analysis of existing models is promising and could lead to new universal retrieval methods. However, the current proposed method seems to be a direct application of existing methods in the literature. The introduction and related work could be enhanced to clearly showcase any technical contributions. The proposed work could be further improved by comparing the task-sensitivity adaptive learning rate method with other adaptive learning methods in the literature.

---

> ### Author Response · Authors · 2022-11-18
> **Author Response**
>
> We would like to thank the reviewer for the comments.
>
> Performance:
> - Improvement is not marginal. The average R-precision of TACO-DR compared to the next best baseline is:\
> 73.74 vs 71.38 (page-level)
> 50.80 vs 48.07 (passage level)
>
> - We emphasize that per-task ANCE (the next best baseline) consists of 8 separate models optimized for individual tasks and should be considered as having an unfair advantage over TACO-DR which is a single model. Even so, TACO-DR significantly outperforms per-task ANCE in average precision.
>
> - We also note that MT-CorpusBrain (concurrent with our submission) is a generative retriever, not a dual encoder. It comes with its own limitations and requirements that we discuss in the paper. We only list their results for reference not for comparison, even though we beat them (73.74 vs 72.56).
>
>
> Novelty:
> - Our work is not a straightforward application of adaptive learning rates. We are not addressing generic gradient-based optimization that existing momentum-based optimizers such as Adagrad and Adam are concerned with. We are addressing multi-task optimization. Our solution is a novel adaptive update for encouraging task specialization in parameters. Our method can be used in conjunction with such generic adaptive optimizers. We are considering a different problem in the optimization landscape. Indeed our optimization is built on top of ADAM with the emphasis on learning task-specific signals.
>
> - Our work is not a straightforward application of prompt tuning. In fact, prompt-based approaches were found to be unhelpful in previous works (see Maillard et al.). We show that with a careful choice of task identification prompts and proper model optimization, we can achieve significant improvements. This finding is important and not an artifact of just applying prompt tuning.
>
> More baselines:
> - We have added GradNorm as another loss-level adaptive update baseline in addition to CGD. Both CGD and GradNorm are instances of the “self-paced version” that the reviewer suggests. The “entropy-based task-specific learning rate” that the reviewer suggests IS a key building stone of our work, while we move further with more fine-grained measurements from sensitivity. Ours is the only method that updates parameters adaptively both at the parameter and at the task level. Please see the general response.
>
> Response to specific questions and comments:
>  - Special Fraction in Fig 3(b) is: the fraction of the number of task-special parameters among all the parameters. Task-special parameter means the parameter has entropy <= 0.3.
> - Ablation study actually shows the proposed model outperforms MT-T5-ANCE substantially (3.13% improvement): 73.74% vs 70.61% average page-level R-Precision.
> - Task commitment is a fraction of task-specific parameters that are committed to the same task during K training steps, i.e., being most sensitive to the same task. It can reflect the usefulness of momentum. Comparing task commitment for different models doesn’t make sense.

---

> > ### Comment · Reviewer_KDMG · 2022-11-24
> > **Post-rebuttal**
> >
> > Thank you for the responses that have addressed some of my concerns. I am increasing my rating to reflect this but have a couple of additional comments:
> >
> > I am wondering if a relative gain of 2% is statistically significant [73.74 vs 71.38 (page-level) and 50.80 vs 48.07 (passage level)]. Please remind me if the results are averaged over multiple experimental runs, as I cannot seem to locate this information in the paper.
> >
> > In my view comparing task commitment across models could show the task sensitivity differences across models and this seems to somewhat correspond to the effective number of [task-specific in this case] parameters.

---

### Official Review · Reviewer_ZTG9 · 2022-10-25

**Confidence:** 2
**Correctness:** 4
**Technical Novelty And Significance:** 3
**Empirical Novelty And Significance:** 4
**Recommendation:** 6

**Clarity, Quality, Novelty And Reproducibility:**

The paper is clear and the idea seems novel to me (but I'm not an expert in this area).

**Strength And Weaknesses:**

The paper is generally well-written and easy to follow. I have the following comments and questions:
- If I understand correctly, the results in table 2 of the paper should be comparable to table 3 of Chen et al. (2022). It seems that their best results are slightly different from what is reported in table 2. For instance, for T-REx and TQA Chen et al. (2022) are reporting 85.03 and 71.71 while the results in table 2 are 77.62 and 68.78. If possible and in order to make the results more comparable, can you use a similar training strategy to theirs?
- For the sake of comparing the models on all 11 KILT tasks, can you also add the results for the 3 remaining tasks?
- Can you provide the confidence intervals in Figure 3(a)

**Summary Of The Paper:**

The paper studies the performance of multi-task learning in universal dense retrieval systems. The authors show that the network capacity is not the limiting factor for the universal retrieval accuracy. Instead due to insufficient optimization, large portion of parameters show low sensitivity to each task. A method, called TACO-DR, is proposed that optimizes the task-specialty of the parameters.

**Summary Of The Review:**

Overall, I found the idea of having task-specific adaptive learning rate interesting. However, I believe the empirical results can be further improved.

---

> ### Author Response · Authors · 2022-11-18
> **Author Response**
>
> We would like to thank the reviewer for the comments.
> - Corpus-brain (chen et al. 2022) : We compare with their MT baseline, which is the fair comparison. Their final model uses additional data and has a pretraining stage, which is not a fair comparison with models only using the KILT dataset. Corpus-brain is a generative retriever, not a dense retriever, which means it has different trade-offs as we discussed in the paper. We list their results for reference, not for comparison, even though we beat them (73.74 vs 72.56).
> - We don’t plan to add the remaining 3 KILT datasets results because Corpus-brain is not our main baseline. We only list their results for reference (not for comparison). We mainly compare with MT-DPR paper baselines since we focus on dense retrieval instead of generative retrieval. We can’t use a similar training strategy as Corpus-brain because dense retrieval and generative retrieval are completely different techniques.
> - In our experiments we do not observe much variance in random runs on Figure 3.a. The model performs quite stable even with different hyperparameter values—they are more stable for different random seeds. We are happy to add them if you think they are important in the next version.

---

### Author Response · Authors · 2022-11-18
**General Response**

- General Response:

We would like to thank the reviewers for feedbacks and comments. We have added GradNorm as another loss-level weighting baseline besides CGD. Note that our method is parameter-level instead of loss-level like GradNorm or CGD. GradNorm or CGD doesn’t consider the parameter dimension and uses the same weight/learning rate for all the parameters.

More specifically, let $g = \sum_t g_t$ denote the gradient of the multi-task objective where $g_t$ is the gradient of the $t$-th task loss. The loss-level adaptive update such as GradNorm and CGD has the form

$ g = \sum_t w_t g_t  $

where $w_t$ is the weight for task $t$. In contrast, our adaptive update has the form

$g(i) = \sum_t w_t(i)  g_t(i)$

where $u(i)$ denotes the $i$-th element of vector $u$. Our update is more fine-grained since we have a weight for each task and parameter rather than just for each task.


- Updated Results:

**GradNorm Results**:

|                  | FEV | AY2 | T-REx | zsRE | NQ | HoPo |  TQA |  WoW | AVG|
|:------:|:------:|:----------:|:-----------:|:---------:|:---------------------:|:---------------------:|:---------------------:|:---------------------:|:---------------------:|
| no prompt |  84.1/57.39 | 85.18 | 70.08/64.45 | 84.94/68.61 |  62.71/29.43   |  49.09/43.24 |  64.43/25.09  |   60.05/32.68  |  70.07/45.84  |
| prompt |  84.7/58.3 | 85.28 | 75.32/69.6 | 91.73/75.54 | 63.8/32.45  |  51.97/47.7 |  69.3/29  |  60.31/34.18  |  72.80/49.54  |
| TACO-DR| 86.17/60.76| 84.64 | 78.12/72.57| 97.91/82.80| 61.86/31.16| 50.61/46.72 | 69.62/28.32 | 60.97/33.24 | 73.74/50.80|


As expected, GradNorm performs similarly with more recent multi-task learning methods (PCG: Gradient Surgery and CGD: CommonGood) we compared and does not outperform single-task models. TACO-DR remains necessary to obtain better multi-task performance than task-specific models.

---

### Decision · Program_Chairs · 2023-01-20

**Decision:**

Reject

**Justification For Why Not Higher Score:**

The submission is not of ICLR quality.

**Justification For Why Not Lower Score:**

The submission is not of ICLR quality.

**Metareview: Summary, Strengths And Weaknesses:**

Strength
* Techniques for improving multi-task learning in universal retrieval are proposed.
* Experiments are conducted.

Weakness
* The novelty is limited. The method is a combination of existing techniques.
* The experimental results need to be more convincing. The work is not significant enough.
* There are places that need to be better explained.

The authors have added one more baseline during the rebuttal. However, the concerns on the novelty and significance remain. The authors have clarified some of the problems. However, it is unclear whether they can fix all the issues if the submission is accepted.

**Summary Of Ac-Reviewer Meeting:**

I initiated a discussion. One reviewer replied and supported rejection.